# Symptomatic Pneumorrhachis from Bronchial-Subarachnoid Fistula

**DOI:** 10.3390/diagnostics14192170

**Published:** 2024-09-29

**Authors:** Alexander W. Lipinski, Mathew V. Smith, Eric J. Wannamaker, Vincent M. Timpone

**Affiliations:** 1Mayo Clinic Alix School of Medicine, Arizona Campus, Mayo Clinic College of Medicine and Science, Scottsdale, AZ 85259, USA; 2Department of Radiology, Mayo Clinic, Phoenix, AZ 85054, USA; smith.mathew@mayo.edu (M.V.S.); wannamaker.eric@mayo.edu (E.J.W.); timpone.vincent@mayo.edu (V.M.T.)

**Keywords:** pneumorrhachis, bronchial-subarachnoid fistula, post treatment changes, pre-operative imaging, post-operative monitoring

## Abstract

Bronchial-subarachnoid fistulas are rare occurrences, which are not well defined in the literature. This uncommon clinical phenomenon may result in symptomatic pneumorrhachis and presents unique clinical challenges. This report details a case of a 53-year-old female whose treatment for recurrent chondrosarcoma of the thoracic spine included multiple surgeries and radiotherapy. Two weeks after her most recent debulking surgery, she experienced a rapid onset of unusual symptoms, including headache, back and neck spasms, bladder incontinence, and confusion. The source of her symptoms was found to be secondary to pneumorrhachis from a pre-existing bronchial-pleural fistula that had fistulized to the subarachnoid space discovered on computed tomography (CT) and confirmed intraoperatively. The patient was treated successfully with high-flow oxygen therapy and bed rest, followed by surgical correction of both a pleural air leak and a dural defect with muscular flaps. The patient was discharged home in stable condition and remained clinically free of recurrent bronchial-subarachnoid fistula six months after surgical repair. This case contributes to the existing literature by providing detailed clinical insights into the diagnosis and successful management of a bronchial-subarachnoid fistula leading to pneumorrhachis, thereby highlighting the importance of early recognition and intervention and underscoring the need for further research in this area.

**Figure 1 diagnostics-14-02170-f001:**
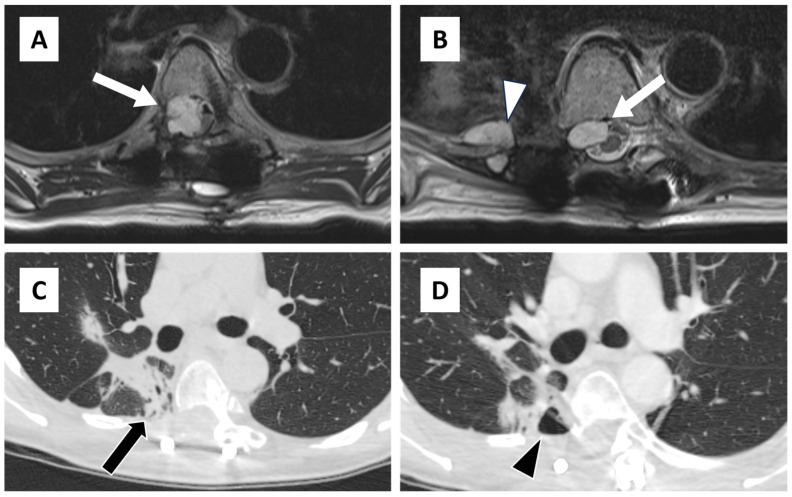
Axial T2-weighted fast spin echo images taken 1 month prior to repeat chondrosarcoma resection demonstrate recurrent T2 hyperintense tumor in the epidural space (arrow) above the prior operative bed (**A**) as well as within the prior operative bed (**B**) with additional tumor along the proximal edge of the partially resected right 8th rib (arrowhead). Axial pre-procedural planning CT of the thorax (**C**) demonstrates post radiation changes in the adjacent lung with focal bronchiectatic consolidation in the medial right lower lobe (arrow) and (**D**) a small adjacent contained hydropneumothorax (arrowhead). There was no specific evidence of pleural damage during the debulking surgery; however, the procedure was notably challenging due to altered anatomy from prior surgeries and radiation therapy. The patient had a chronic dry cough that did not worsen after surgery, and there was no cerebrospinal fluid expectoration observed. CT scans of her chest consistently showed no other significant pneumothorax or pleural effusion.

**Figure 2 diagnostics-14-02170-f002:**
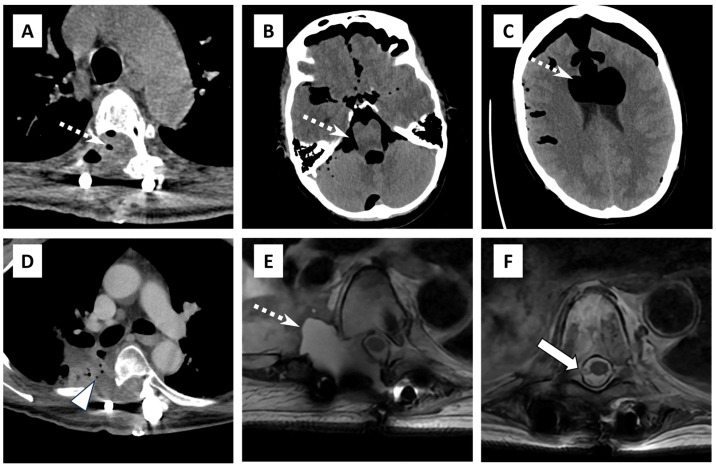
Computed tomographic (CT) images of the thoracic spine (**A**) and head (**B**,**C**) 2 weeks after the most recent repeat resection demonstrate pneumorrhachis (dashed white arrow in (**A**)) and secondary pneumocephalus (dashed white arrows in (**B**,**C**)). Contrast-enhanced chest CT (**D**) taken at the same time as (**A**–**C**) demonstrates an organized fluid collection in the surgical bed extending into the pleural space laterally, abutting bronchiectatic airways (arrowhead). Axial T2-weighted fast spin echo from same day magnetic resonance imaging (**E**,**F**) redemonstrates the perioperative fluid collection (dashed white arrow in (**E**)) and reveals a dural defect in the right lateral thecal sac at the resection level (white arrow in (**F**)). Pneumorrhachis is primarily diagnosed via CT scan, which is the gold standard due to its high sensitivity in detecting air within the spinal canal, however comparative studies on the efficacy of MRI are lacking. MRI is used as a secondary tool to assess associated soft tissue injuries or rule out other conditions like spinal cord compression [1,2,3].

Current literature on bronchial-subarachnoid fistulas and pneumorrhachis is notably sparse, primarily consisting of isolated case reports. Risk factors for developing these fistulous communications include trauma, prior surgery, infection, chronic inflammatory conditions, and prior radiation therapy [4,5]. Of the few reported cases of pneumorrhachis related to bronchial-subarachnoid fistula, the condition represents a delayed post-treatment complication, with earlier reports documenting symptom onset ranging from two months to ten years postoperatively [6,7,8]. The clinical manifestations of pneumorrhachis and secondary pneumocephalus are variable, including myelopathy, radicular pain, neck pain, and headaches, and may be due to the pressure and/or inflammatory effects of intrathecal and intracranial gas [4].

**Figure 3 diagnostics-14-02170-f003:**
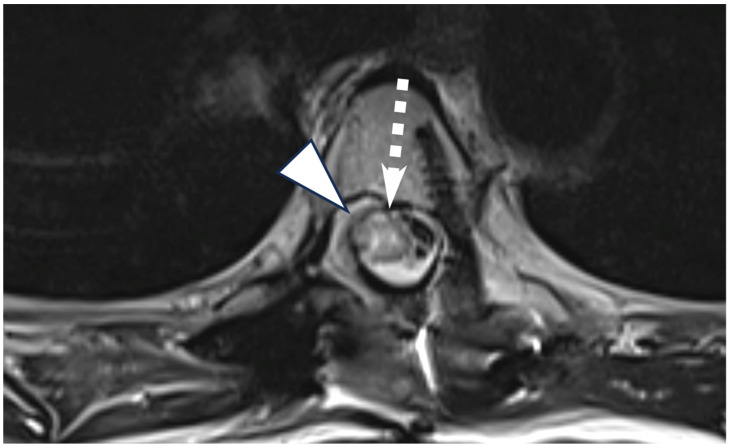
Axial T2-weighted turbo spin echo imaging performed after subsequent exploration and flap reconstruction demonstrates a repaired dural defect (arrowhead) and resolved pseudomeningocele with some residual myelopathic signal changes (dashed arrow). Prompt recognition of pneumorrhachis secondary to bronchial-subarachnoid fistulas and the implementation of urgent treatment strategies to mitigate serious neurological complications are essential. These strategies include optimizing patient positioning to isolate gas collections to safer locations and operative planning to seal off the underlying fistula [4,6,9,10]. Conservative management and monitoring are the first-line for pneumorrhachis and are effective in preventing neurological complications, with most cases resolving spontaneously [11]. Hyperbaric oxygen therapy (HBOT) is considered a potential treatment modality for symptomatic pneumorrhachis and works by increasing the partial pressure of oxygen, which accelerates the reabsorption of intrathecal air and promotes nitrogen washout from air collections [1]. Additionally, HBOT has been found to significantly reduce the volume of pneumocephalus more rapidly compared to normobaric oxygen therapy, leading to faster clinical improvement [12].

Surgical intervention is typically reserved for severe neurological symptoms, such as spinal cord compression or CSF leaks, but can improve muscle strength and neurological function in pneumorrhachis cases [13,14]. The positioning of the patient during and after surgery plays a crucial role in both the development and management of pneumocephalus and pneumorrhachis, highlighting the importance of careful intraoperative and postoperative care. Moreover, the development of fluctuating hemiparesis in one case provides insight into the potential neurologic impacts of these conditions, suggesting that air within the spinal canal can affect motor functions in a position-dependent manner [4].

## Data Availability

This article does not include any additional primary data besides the information already presented in the case report section.

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
