# Peer review of "Symptomatic Pneumorrhachis from Bronchial-Subarachnoid Fistula"

_diagnostics, 2024, doi:10.3390/diagnostics14192170_

Round 1

Reviewer 1 Report

Comments and Suggestions for Authors

Actually, CT and MRI images of  seem to be very interested, however, I have some questions  

>line 46, p#2 Figure 2 should be figure 3.

> from diagnostic prospective, how is pneumorrhachis diagnosed, and what imaging modalities are most effective in identifying this condition? 

>What urgent treatment strategies are recommended for managing pneumorrhachis, and how effective are they in preventing neurological complications?

>Can you elaborate on the role of hyperbaric oxygen therapy in the management of pneumorrhachis?

>What gaps exist in the current literature regarding bronchial-subarachnoid fistulas and pneumorrhachis, and how does this case contribute to addressing those gaps?

Author Response

Thank you very much for taking the time to review our manuscript. We appreciate your insightful comments and suggestions, which have helped us improve the quality of our work. Please find the detailed responses below, and note that all corresponding revisions and corrections are highlighted in blue in the resubmitted manuscript.

Comments 1:

line 46, p#2 Figure 2 should be figure 3.

Response 1:

Thank you for pointing this out. We have corrected the figure numbers.

Comments 2:

From diagnostic prospective, how is pneumorrhachis diagnosed, and what imaging modalities are most effective in identifying this condition?

Response 2:

We appreciate this important question. To address it, we have added the following section to the manuscript, highlighted in blue:

"Pneumorrhachis is primarily diagnosed via CT scan, which is the gold standard due to its high sensitivity in detecting air within the spinal canal. However, comparative studies on the efficacy of MRI are lacking. MRI is used as a secondary tool to assess associated soft tissue injuries or rule out other conditions like spinal cord compression [1-3]."

This addition can be found under figure 2.

Comments 3:

What urgent treatment strategies are recommended for managing pneumorrhachis, and how effective are they in preventing neurological complications?

Response 3:

Thank you for raising this crucial aspect. We have included the following information in the manuscript to address your query, highlighted in blue:

"Conservative management and monitoring are the first-line for pneumorrhachis and are effective in preventing neurological complications, with most cases resolving spontaneously [11]. Surgical intervention is typically reserved for severe neurological symptoms, such as spinal cord compression or CSF leaks, but can improve muscle strength and neurological function in pneumorrhachis cases [13,14]."

This revision is located under figure 3.

Comments 4:

Can you elaborate on the role of hyperbaric oxygen therapy in the management of pneumorrhachis?

Response 4:

We appreciate the opportunity to elaborate on this treatment modality. The manuscript has been revised to include the following details, highlighted in blue:

"Hyperbaric oxygen therapy (HBOT) is considered a potential treatment modality for symptomatic pneumorrhachis and works by increasing the partial pressure of oxygen, which accelerates the reabsorption of intrathecal air and promotes nitrogen washout from air collections [1]. Additionally, HBOT has been found to significantly reduce the volume of pneumocephalus more rapidly compared to normobaric oxygen therapy, leading to faster clinical improvement [12]."

This elaboration can be found under figure 3.

Comments 5:

What gaps exist in the current literature regarding bronchial-subarachnoid fistulas and pneumorrhachis, and how does this case contribute to addressing those gaps?

Response 5:

Thank you for highlighting this important area. We have expanded the manuscript to discuss the existing gaps and the contribution of our case as follows, highlighted in blue:

Abstract:

"This case contributes to the existing literature by providing detailed clinical insights into the diagnosis and successful management of a bronchial-subarachnoid fistula leading to pneumorrhachis, thereby highlighting the importance of early recognition and intervention, and underscores the need for further research in this area."

Figure 2: Pneumorrhachis is primarily diagnosed via CT scan, which is the gold standard due to its high sensitivity in detecting air within the spinal canal, however comparative studies on the efficacy of MRI are lacking.

Figure 3: Current literature on bronchial-subarachnoid fistulas and pneumorrhachis is notably sparse, primarily consisting of isolated case reports.

We believe that the revisions made adequately address all the reviewer’s comments and enhance the clarity and depth of our manuscript. We are grateful for the constructive feedback and are confident that the changes have strengthened our work.

Thank you once again for your valuable time and consideration.

Sincerely,

Alexander W. Lipinski

Reviewer 2 Report

Comments and Suggestions for Authors

The manuscript shown a rare and interesting imaging about the pneumorrhachis from bronchial-subarachnoid fistula. The manuscript could be accepted after minor revision.

1, Was there any damage to the pleura during the surgical procedure?

2, The presence of cerebrospinal fluid leakage in the thoracic vertebrae did not result in significant fluid pneumothorax, whether the patient has a cough or even cerebrospinal fluid expectoration?

Author Response

Thank you very much for taking the time to review our manuscript. We appreciate your positive feedback and constructive comments, which have helped us improve the quality of our work. Please find the detailed responses below, and note that all corresponding revisions and corrections are highlighted in blue in the resubmitted manuscript.

Comments 1:

Was there any damage to the pleura during the surgical procedure?

Response 1:

Thank you for your question. We have added the following information under Figure 1 to address this concern, highlighted in yellow:

"There was no specific evidence of pleural damage during the debulking surgery, however the procedure was notably challenging due to altered anatomy from prior surgeries and radiation therapy. The patient had a chronic dry cough that did not worsen after surgery, and there was no cerebrospinal fluid expectoration observed. CT scans of her chest consistently showed no other significant pneumothorax or pleural effusion."

This addition can be found under Figure 1 caption of the revised manuscript.

Comments 2:

The presence of cerebrospinal fluid leakage in the thoracic vertebrae did not result in significant fluid pneumothorax, whether the patient has a cough or even cerebrospinal fluid expectoration?

Response 2:

We appreciate this insightful observation. To address this, we have included the following details in the manuscript under Figure 1, highlighted in yellow:

"The patient had a chronic dry cough that did not worsen after surgery, and there was no cerebrospinal fluid expectoration observed. CT scans of her chest consistently showed no other significant pneumothorax or pleural effusion."

This addition is located under the Figure 1 caption of the revised manuscript.

We believe that the revisions made adequately address all the reviewer’s comments and enhance the clarity and depth of our manuscript. We are grateful for the constructive feedback and are confident that the changes have strengthened our work.

Thank you once again for your valuable time and consideration.

Sincerely,

Alexander W. Lipinski